# Design Strategies for and Stability of mRNA–Lipid Nanoparticle COVID-19 Vaccines

**DOI:** 10.3390/polym14194195

**Published:** 2022-10-06

**Authors:** Ting Liu, Yang Tian, Aiping Zheng, Chunying Cui

**Affiliations:** 1Department of Pharmaceutics, School of Pharmaceutical Sciences, Capital Medical University, Beijing 100069, China; 2State Key Laboratory of Toxicology and Medical Countermeasures, Institute of Pharmacology and Toxicology, Beijing 100850, China

**Keywords:** COVID-19, mRNA, lipid nanoparticle, preparation, quality control, stability

## Abstract

Messenger RNA (mRNA) vaccines have shown great preventive potential in response to the novel coronavirus (COVID-19) pandemic. The lipid nanoparticle (LNP), as a non-viral vector with good safety and potency factors, is applied to mRNA delivery in the clinic. Among the recently FDA-approved SARS-CoV-2 mRNA vaccines, lipid-based nanoparticles have been shown to be well-suited to antigen presentation and enhanced immune stimulation to elicit potent humoral and cellular immune responses. However, a design strategy for optimal mRNA-LNP vaccines has not been fully elaborated. In this review, we comprehensively and systematically discuss the research strategies for mRNA-LNP vaccines against COVID-19, including antigen and lipid carrier selection, vaccine preparation, quality control, and stability. Meanwhile, we also discuss the potential development directions for mRNA–LNP vaccines in the future. We also conduct an in-depth review of those technologies and scientific insights in regard to the mRNA-LNP field.

## 1. Introduction

The unprecedented impact on public health and economic development of severe acute respiratory syndrome coronavirus 2 (SARS-CoV-2 or COVID-19) was evident in December 2019. There were approximately 400 million confirmed cases and a mortality rate of about 1.4% in February 2022. Vaccines have been widely recognized by the public for their success and effectiveness in preventing disease. Moreover, various forms of therapy and prophylactic vaccines, including peptides, RNA-, viral vector-, DNA-, and protein-based vaccines, have been developed since the outbreak of COVID-19 [1,2]. The most promising vaccine candidates have been transited from viral vector- and protein-based technologies into messenger RNA (mRNA) and lipid nanoparticle (LNP) technologies. The mRNA vaccine consists of mRNA encoding a particular antigenic protein into the human body, directly translated to form the corresponding antigen protein, thereby eliciting a potent immune system response to produce antibodies against diseases [3,4]. One of the key advantages of mRNA vaccines is that they can be designed and manufactured within a short time scale to meet human requirements, which makes them very appropriate for responding to the pandemic. It is essential for the effective delivery of mRNA in vivo to achieve preventive effects. However, exogenous mRNA must cross the barrier of the host cell membrane into the cytoplasm and translate into immunogenic proteins [5]. The mRNA vaccine is quite unstable and requires the assistance of some vectors in the cell membrane of the host cells. Additionally, the sensitivity of mRNA vaccines to RNA enzyme-catalyzed hydrolysis pose significant challenges to their intracellular delivery [6,7]. LNPs are, clinically, one of the most advanced mRNA carriers that can load and deliver mRNA to cells while avoiding mRNA degradation [8,9]. After the mRNA is delivered to a cell, it instructs the cell to produce protein, thereby eliciting an immune response. Therefore, using mRNA, LNP, and delivery system technologies achieve a desired biological response. To date, the research conducted on mRNA–LNP vaccines used to combat viral diseases, including Ebola, Zika, HIV, influenza, and cytomegalovirus infections, has been performed in many countries. Table 1 summarizes some of the applications of RNA-loaded LNPs that have been reported in the literature. It is worth emphasizing that a COVID-19 vaccine with LNPs used as a delivery vehicle developed by BioNTech/Pfizer and Moderna has been granted emergency authorization from the U.S. Food and Drug Administration (FDA) (Table 2) [10,11,12].

In the current review, we briefly describe the key stages in the preclinical development of mRNA–LNP vaccines, including a selection of immunogens and delivery vehicles, the preparation of mRNA–LNP vaccines by a microfluidic method, quality control, and the stability of mRNA–LNP vaccines. Simultaneously, we also discuss the existing problems of LNPs in mRNA formulations and present the possible directions for the research in the future.

## 2. The Design Strategies for mRNA–LNP Vaccines

### 2.1. Selection of Immunogens

The immunogen plays an irreplaceable role in the process of vaccine development. The ideal immunogen can produce a persistent humoral and cellular immune response via stimulating the host cell [25]. SARS-CoV-2 is a novel β-coronavirus with a genome size of approximately 30 kb [26,27,28,29]. That is, no fewer than 84,000 complete genomes in SARS-CoV-2 sequences (>29 kb) have been recorded in the Global Initiative of Sharing All Influenza Data (GISAID) database [29,30]. The high volume of gene sequencing data provide a great opportunity to develop vaccines and understand the trend of virus evolution.

When imaged by transmission electron microscopy (TEM), it can be observed that SARS-CoV-2 is spherical in shape with a particle size of 70–130 nanometers (Figure 1A). The four major structural proteins of SARS-CoV-2 are as follows (Figure 1B): spike protein (S), envelope protein (E), membrane glycoprotein (M), and nucleocapsid protein (N). In addition, SARS-CoV-2 contains a positive-sense, single-stranded RNA genome related to nucleoproteins [30,31]. The S protein is a homologous trimer distributed on the surface of COVID-19 that mediates it into cells during the infection process [32,33]. Additionally, the S protein consists of S1 and S2 subunits [27,34]. The S1 subunit is located outside the envelope of the virus, which is mainly combined with vascular angiotensin-converting enzyme 2 (ACE2) [25,35,36], and the latter is the transmembrane subunit that can mediate the fusion of viral and cellular membranes [37]. The S1 region contains the receptor-binding domain (RBD), whereas the S2 subunit contains a putative fusion peptide, transmembrane domain, and two heptad repeat regions called heptad repeats 1 and 2 (HR1 and HR2) [38,39,40,41]. During the early stages of virus infection, the S protein plays a major role and effectively induces the reaction of T cells in the body [32]. Recently, substantial evidence proved that the SARS-CoV-2 infection could stimulate the production of S-protein-specific CD4+ and CD8+ T cells [42,43]. The special structure of the S protein provides a potential therapeutic target for the development of broad-spectrum anti-coronavirus drugs. The S protein regulates its exposure to receptor binding sites and subsequently undergoes structural rearrangements to drive fusion between the virus and the cell membrane [27,44]. Therefore, many vaccines in development, at present, target the S protein as the main antigen.

The N protein, the most abundant protein in virions, plays a vital role in the encapsidation of viral genomic and antigenomic RNA in a ribonucleoprotein complex [45]. Moreover, N proteins have an essential role in viral genomic RNA replication and mRNA transcription [46,47]. Furthermore, they are involved in the immune-regulation process [48]. N proteins can stimulate a strong TH2 response and induce tissue damage, for which it may be difficult to provide effective protection [49]. The E protein is only a minor membrane component, but nevertheless has an important effect on in coronavirus assembly [50]. Structural M proteins mainly mediate the assembly and release of enveloped viral particles, and exert an important role in the structural stability and functional expression of other proteins [51]. Thus, the other structural proteins of SARS-CoV-2 are not deemed as an effective vaccine immunogen for clinical study.

### 2.2. Selection of Delivery Vehicle

mRNA presents great potential in the application of viral vaccines, cancer immunotherapy, and genome editing [4,52,53]. Nucleic acid is highly susceptible to degradation by several nucleases in vivo [3], so it is required for the development of effective and safe drug delivery systems (DDSs) to reach specific target cells and produce potent and predictable effects. At present, different types of delivery methods have been commonly exploited as DDSs for nucleic acids (Figure 2) [54]. Particular attention has been paid to designing lipid-based mRNA DDSs that protect nucleic acids and ensure their targeted delivery, controlled release, and enhanced cellular uptake.

#### 2.2.1. Lipid Nanoparticles

Nanoparticles have a wide range of biomedical applications due to their advanced technology [56]. In particular, LNPs are among the most commonly used mRNA delivery vehicles in both preclinical and clinical studies [57,58,59]. LNPs may present a micelle-like structure that encapsulates nucleic acids inside a non-aqueous core, rather than a lipid bilayer enclosing an aqueous core in traditional liposomes [60,61,62]. Both mRNA and cell membranes are negatively charged, so mRNA cannot cross the cell membrane alone for protein expression. On the contrary, positively charged LNPs have a well-complexed system with the nucleic acid content under the influence of electrostatic action [60,63,64]. Notably, LNPs were initially employed to develop siRNA delivery. Subsequently, the LNP formulation was used to deliver mRNA for cancer immunotherapy in 2014 [65]. In comparison to other drug carriers, LNPs present numerous advantages [12,60,66,67]. For example, LNPs can effectively encapsulate mRNA for effective and efficient delivery into a cell [68]. Moreover, LNPs consists of biocompatible materials suitable for human use, which can be synthesized at a large scale under the GMP level. More importantly, LNPs are robustly synthesized, and their components can be easily adjusted to increase cellular uptake and reduce cellular toxicity [12].

LNPs are mainly composed of a cationic/ionizable lipid and helper lipids, such as phospholipids, cholesterol, and/or poly(ethylene glycol) (PEG) lipids [12,60,69,70]. The structures of lipids commonly used for mRNA delivery are presented in Figure 3 [71], including preclinical studies, clinical stages and authorized lipids. In the context of the LNP system, the cationic/ionizable lipid is a critical component of LNPs. There is a head group with positive charges in cationic/ionizable lipids, which serves as two functions [23,72]: (i) increasing the cellular uptake and endosomal escape of the LNP system by disrupting the cell membrane; and (ii) improving the mRNA stability and encapsulation efficiency of LNPs by forming electrostatic interactions with the negatively charged mRNA. The ionizable lipids are positively charged at a low pH level, but they remain neutral at a physiological pH [72]. Therefore, it is beneficial for mRNA delivery to mediate passive entrapment and ensure a more biocompatible delivery system. The utilization of ionizable cationic lipids provides a solution for the biocompatibility issues of these systems [60]. Phospholipids and cholesterol contribute to the promotion of the stability of LNPs and endosomal escape by modulating membrane integrity and rigidity [23]. The research showed that cholesterol analogs with C-24 alkyl phytosterols in LNPs can enhance the intracellular delivery of mRNA in vivo [73]. The PEG-lipid is used to control particle size, zeta potential, and reduce nonspecific interactions to prevent aggregation under storage [74]. In terms of marketed RNA vaccines, the PEG-lipids used in Moderna and Biontech’s mRNA vaccines are PEG-2000 conjugates. mRNA–LNP formulations containing PEG2000-DMG have a higher delivery efficacy in vivo than formulations containing ALC-0159 [75]. Moreover, the separation of PEG2000-DMG from LNPs is quicker than that of ALC-0159, thereby facilitating the cellular uptake and endosomal escape of LNPs. Lipid molecules of different compositions can wrap mRNA molecules inside them to form a spherical LNP, and LNPs with different particle sizes can be prepared by adjusting the prescription, as presented in Figure 4. Therefore, to develop an effective mRNA vaccine for COVID-19, the LNPs can be appropriately designed and adjusted to obtain mRNA based on immunotherapies with high immunogenicity and good therapeutic outcomes.

Membrane lipids are essential for protein trafficking and aggregation, membrane fusion, and signal transduction. Lipidomic studies have identified a wide variety of lipids in membranes. Changes in lipid organization can have effects on cellular functions, such as signal transduction and membrane trafficking [77,78]. Cholesterol is one of the most important regulators for lipid organization. Due to its unique mechanism, the body can maintain the level of cholesterol in the cell membrane within a suitable range [79]. The different types of lipid organization have important effects on membrane proteins. Many membrane proteins are more likely to be associated with specific types of lipid organization, which would lead to its physical separation in a bilayer with a lipid phase coexistence [80,81]. When genes or the environment change, the orderly arrangement of lipid organization becomes disordered; these membrane effects may cause discomfort in the human body.

LNPs are nanoparticles formed by lipid components. When LNPs load genetic material into cells, the degree of impact on the morphology and function of the lipid organization is essential for the composition and dosage of the selection of LNPs. The development of more efficient and safer LNPs for intracellular mRNA delivery is a long-standing challenge. LNPs are one of the most effective means of mRNA delivery. We can select an appropriate lipid composition or modify its structure to improve its biodegradability in vivo, so as to improve biocompatibility, avoid the toxicity and side effects caused by lipid accumulation. At present, a series of studies mainly focus on the linker fragments of cationic lipids (ionizable lipids), such as esters, amides, and mercaptans [82]. Dlin-mc3-dma, ALC-0315, and SM-102 all contain ester-bond linker fragments. Lipids linking degradable fragments are often rapidly cleared from the body and can be used in multiple doses with fewer side effects. Dlin-MC3-DMA is an FDA-approved excipient. In order to further improve the therapeutic index of siRNA-LNP, its biodegradable version, L319, was develop by adding ester bonds to the hydrophobic dialkyl chains, which enables the rapid elimination of siRNA-LNPs in the liver [83]. ATX lipids (Arcturus) were also designed to contain an ionizable amino head group and a biodegradable lipid backbone, which can degrade and scavenge in the liver much faster than Dlin-MC3-DMA [84]. Embedding biodegradable linker fragments in lipids may improve their biocompatibility and further promote their elimination in vivo [83]. Therefore, it is important to develop new next-generation lipids with a superior potency and biodegradable functions to further promote the development of the LNP platform.

The main advantages of LNPs as DDSs include the tailored optimization of biophysical and biological characteristics to achieve efficient encapsulation and controlled release. The current review focused on the research overview of LNPs as a mRNA DDS.

#### 2.2.2. Other Types of Lipid-Based mRNA DDSs

In addition to the LNPs previously mentioned, other types of lipid-based mRNA DDSs include hybrid biopolymers, lipoplexes (LPXs), lipopolyplexes (LPPs), and nanocapsules (LNCs). Recently, lipid polymer hybrid nanoparticles (LPNs) have served as promising biomaterials for RNA delivery that integrate the physicochemical properties of both lipids and the polymers. The LPN can effectively protect and deliver mRNA to achieve the correct expression of the corresponding protein. A study has shown that polyβ-amino lipids (PBAEs) were coated with phospholipid bilayers, and then the mRNA was loaded onto the surface of the nanoparticles by an electrostatic interaction to deliver mRNA vaccines through nasal mucosa [85]. The results indicate that the LPN/mRNA complex is readily taken up by dendritic cells (DC_S_) and can deliver mRNA into the cytoplasm of DCs to reduce toxicity [85]. In addition, Ball et al. combined mRNA and siRNA into a polymer composed of lipids modified with ionizable lipids, cholesterol, and polyethylene glycol to investigate the co-delivery effect of mRNA and siRNA. The results show that siRNA and mRNA in the same LPN enhanced the drug efficacy both in vivo and in vitro [86]. Using LPXs and LPPs to efficiently deliver nucleic acid has attracted the attention of many researchers. LPXs and LPPs can deliver genes to dendritic cells, which can better activate the immune response of T cells through antigen presentation, achieving the ideal immunotherapy effect. At present, many cationic LPXs are used, such as polyethylene imine, poly (β-amino ester), PBAE, and chitosan. BNT111 is an mRNA vaccine encoding NYESO-1, tyrosinase (Tyr), MAGE-A3, and TPTE developed by BioNTech, which are delivered intravenously to patients via RNA-LPX. The results showed that BNT111 can induce a specific antitumor immune response with good safety and tolerance [87]. Mockey et al. demonstrated that mice receiving systemic injections of MART1 mRNA histidylated LPPs were specifically and significantly protected against B16F10 melanoma tumor progression [88]. LNCs are submicron particles composed of an oily liquid core surrounded by a solid or semisolid shell. LNCs are biomimetic nanocarriers used for the encapsulation of a wide variety of active ingredients. They are highly stable compared to other nanocarriers, and they have been observed to confer highly desirable characteristics to internally delivered therapeutic molecules [89]. The lipid-based mRNA DDSs mentioned above were not highlighted in the current review.

### 2.3. Preparation of mRNA–LNP Vaccine

The workflow of mRNA–LNP is presented in Figure 5. First, the antigen gene was combined with blank plasmid by PCR technology to form a recombinant plasmid, and then the plasmid containing the target gene was introduced into the cells of E. coli bacterium. In order to obtain a large quantity of cloned genes, multiplying bacteria were cultured in LB liquid medium (37 °C) overnight. Generally, E. coli bacterium are lysed by an alkali to release the plasmids. Subsequently, the extracted plasmids needed to be sequenced to ensure that the gene sequence of the virus was not altered. Following the quality test, enzymes could linearize circular plasmids and separate the antigen genes. The linearized DNA was mixed with nucleoside triphosphate (NTPs: ATP, GTP, UTP, and CTP) using enzymes (helicases) to open the DNA template and transcribe it into mRNA. In addition to the target product, the in vitro transition of mRNA also contains many impurities, such as enzymes, abnormal transcription products, and DNA. The lithium chloride precipitation method is commonly used to purify mRNA, but the disadvantage of this method is that it cannot remove abnormal transcription products. Some studies have shown that the protein yield of mRNA purified by RP-HPLC has increased 10–1000 times [90,91]. In addition to the methods described above, the anion exchange, ultrafiltration, magnetic bead, and dialysis methods can also be used for mRNA purification. Following purification, the mRNA was sequenced again to ensure the accuracy of the results. Finally, the mRNA was encapsulated in the lipid and formed vaccine particles. The final stages of aseptic filtration and packaging were completed after the quality test of the intermediate product was qualified.

In the past, ethanol injection and thin-film hydration were the preparation methods commonly used for obtaining LNP formulations. With the improvement of technology, large-scale production, which proves difficult for obtaining a greater mRNA encapsulation efficiency and uniform particle size, is generally no longer practiced [92,93]. Recently, the preparation of mRNA–LNPs utilizing T-junction- and microfluidic-mixing platforms has become a popular method. In this section, we focused on the preparation of mRNA–LNPs using microfluidic technology. In recent years, microfluidic devices and technologies have been used for the preparation of LNPs. This technology provides many advantages for the production of mRNA-LNPs, including precisely controlling LNP size, high reproducibility rate, and the provision of a continuous LNP production process. To date, a variety of microfluidic devices have been used in the development of nano-delivery platforms, such as RNA, DNA, and LNP. This method is relatively convenient for demand rate and easy to achieve an increase in production. The specific process of preparation is as follows: first, a molar ratio of 50:10:38.5:1.5 (ionizable lipid:helper lipid:cholesterol:PEG-lipid) was dissolved in ethanol [94,95]. Then, all the nucleic acid was dissolved in 20–25 mM of buffer (citrate or sodium acetate buffers), pH 4.0. Of note, the preparation and handling of the nucleic acid solutions were performed under sterile and RNAse-free conditions. Finally, the alcoholic solution of the lipids was mixed with the aqueous solution of the mRNA by utilizing a microfluidic mixing technique.

The microfluidic mixing technique offers several advantages relative to other methods. By using this method together with the mRNA, the components formed particles of approximately 60–100 nm in size [92]. Meanwhile, mRNA–LNPs with a high encapsulation efficiency (approximately 90%) facilitated the manufacturing process of commercial mRNA drugs under GMP [63,96]. Recently, one study indicated that flow rate was negatively associated with the particle size and PDI of mRNA–LNPs [3,97]. However, the increment of the flow rate over 2 mL/min had no effect on the size of the nanoparticles [3]. Therefore, it is necessary to pay attention to the effects of different instrument parameters on the particle size and PDI during the preparation process. Specifically, in order to obtain nanoparticles with a uniform particle size and good polydispersity index (PDI), the flow rates of the aqueous and organic phases were 1.5 mL/min and 0.5 mL/min, respectively [98].

### 2.4. Quality Control of mRNA Vaccines

Most of the methods used in the preparation of mRNA–LNP are complex. Therefore, a perfect quality control system is crucial in determining the quality of the final product. Safety, efficacy, and quality control elements of the vaccine production process are determined by measuring the critical process parameters (CPPs) and intermediate critical quality attributes (IQAs) [99]. The target-gene sequence design, raw material, mRNA purity and integrity, and mRNA/lipid ratio are directly related to the quality of mRNA vaccines. The quality control of mRNA vaccines is mainly reflected in the plasmids, lipids, enzymes, nucleotides, semi-finished products, and final products used in the production process [100]. mRNA molecules and lipid excipients are essential in the process of mRNA vaccine research. Plasmid is the starting material to produce mRNA, the quality of mRNA that determine the potency of vaccine. In the entire quality control process, the biological characteristics of plasmids, such as purity, the proportion of the superhelical structure and residual material extracted from plasmids, E. coli, copy number, and strain purity used in the plasmid transformation process, should also be considered. Lipids are primarily considered in relation to their source, and physical and chemical characteristics (e.g., appearance, type, purity, and residual solvent). Lipids should also be evaluated for their function and compliance to applicable regulations for novel lipid excipients. In addition, the quality of mRNA–LNP intermediates was assessed using key indicators, such as mean particle size, particle-size distribution, mRNA content, encapsulation rate, and sterility. The mRNA vaccine can induce a strong immune response, the global demand is high, and the mRNA vaccine platform is a new form of technology, which encourages a greater demand for improved production technology, production scale, production cycle, and product quality of the mRNA vaccine [101]. Hence, it is essential to establish a set of quality management systems suitable for mRNA vaccines and apply it to the entire production process and life cycle of mRNA vaccines to ensure better product quality.

## 3. The Stability of mRNA–LNP Vaccines

The long-term stability for mRNA–LNP vaccines has always been a concern in terms of preparation and storage. To prevent the growth of the particle size and the loss of bioactivity, many mRNA–LNP formulations are generally stored at −20 °C or in even lower temperatures for several days [102]. Hence, these factors not only increase the cost of transportation, but are difficult to popularize in countries lacking the cold chain. At present, for example, the Moderna mRNA-1273 vaccine is often stored at temperature ranging from −25 °C to −15 °C, and is directly injected into the host following thawing [24,94,103,104]. The Pfizer/BioNTech BNT162b2 vaccine needs to be stored at temperatures ranging from −80 °C to −60 °C, and then thawed and diluted with saline before injection [94,104]. Stability plays a critical role in the quality of mRNA–LNP vaccines. In this section, we discuss the physical and chemical stability of the mRNA vaccine and proposed the strategies to optimize mRNA vaccine stability.

### 3.1. Physical Stability

The physical stability of mRNA–LNP vaccines mainly includes the aggregation, fusion, and leakage of the encapsulated ingredient (Figure 6). The vaccine has poor storage stability, which is prone to some aggregation and fusion [105]. A size increase over time from LNP aggregation has been commonly observed in previous formulations [106]. The main reason for the aforementioned changes is the lipid crystalline type transition from high- to low-energy states, and the increase in the order of lattice. Ayat et al. also reported that LNPs aggregated and fused during storage [107]. In order to stabilize mRNA–LNP vaccines in storage, PEGylated lipids are usually added to the prescription [98,108,109], which can prevent LNPs from aggregating. However, Schöttler S et al. documented that LNPs containing PEGylated lipids exhibit low cellular uptake and transfection rates [110]. As such, this is a critical factor for PEGylated lipids. For example, distearoyl-phosphatidylethanolamine (DSPE)-PEG2000 is connected to the LNP surface to prevent the leakage of the encapsulated ingredient [48]. Notably, a frozen format has also been a favorable method used to retain stability, to date. Another measure is the addition of a stabilizer, such as sucrose or trehalose, which can prevent the fusion and aggregation of LNPs due to the high viscosity of lipids. Study from Rebecca L Ball et al. showed that LNPs accumulate in the freeze–thaw cycle, but the stability is retained when using freezing protectants, trehalose, and sucrose [111]. In the future, we should explore more suitable stabilizers to develop more stable products.

The stability, potential, and other properties of nanoparticles are influenced by the number ratio of positive and negative charges when mRNA–LNPs are synthesized. The positive charge is usually a cationic lipid with ionizable ammonium (N), while the negative charge is a nucleic acid molecule with a high quantity of phosphate (P). These charges can be combined by electrostatic adsorption. The incorrect proportion of both products may lead to some defects, such as a particle size that is too large and poor stability. Therefore, it is very important to correctly calculate the ratio of negative and positive charges (N/P ratio). 

The surface charge of a LNP is also one of the reasons affecting the physical stability of mRNA–LNP vaccines. The surface charge density of particles can affect the interaction between nanoparticles, which controls its stability [112,113]. Uncharged particles or particles with a low charge density aggregate over time, while particles with a higher charge density repel each other, thereby preventing aggregation.

### 3.2. Chemical Stability

At present, chemical stability (lipid and mRNA degradation) is one of the key factors affecting the quality of mRNA–LNP vaccines. Compared to the DNA vaccine, the reason why mRNA–LNP vaccines are susceptible to oxidation/hydrolysis is that they are sensitive to temperature changes and contain lipids with unsaturated acyl chains. Hemolytic phospholipids and fatty acids, as the main degradation products of LNP preparation, not only result in certain toxicity in the body, but also increase the permeability of the drug and lead to the instability of the lipid bilayer [71,114]. Simultaneously, lipid oxidation is a chain reaction initiated by free radicals, preventing or slowing down the formation of hydroxyl radicals, and hydroperoxides can effectively inhibit lipid oxidation. Storage temperature, type of buffer, radiation, and pH are the main factors that affect lipid degradation. Nuclear magnetic resonance (NMR), mass spectrometry (MS), gas chromatography (GC), vibration spectroscopy, liquid chromatography, thin-layer chromatography (TLC), and other methods are commonly used to analyze the chemical structure, molecular weight, and structural changes of lipids [115,116,117,118,119,120].

In addition to the chemical stability of LNPs, the chemical stability of nucleic acids plays an important role in the quality of mRNA vaccines. A review on the stability of nucleic acids stated that the effects of chemical degradation are irreversible, which directly affects the therapeutic effect of the vaccine [121]. Hydrolysis and oxidation are the most common forms of degradation. Oxidation results in the cleavage of bases, strand breaks, and changes in the mRNA secondary structure. Hydrolysis occurs primarily at the phosphodiester bond that forms the main chain of the mRNA molecule [6]. At present, the developed mRNA vaccine is administered by intramuscular injection (IM). The effective delivery and stability of mRNA vaccines in vivo are also affected by the physiological environment [64]. Studies have demonstrated that due to the lack of carrier protection, naked mRNA is susceptible to the destruction of various enzymes in vivo, causing it to rapidly degrade following administration [122]. mRNA is highly sensitive to ribonuclease hydrolases [123]. Therefore, the instability caused by mRNA hydrolysis to vaccine is also worthy of an in-depth study. It is currently unclear how water in the LNP core interacts with mRNA and the extent to which the degradation sites are protected by binding to ionizable cationic lipids.

The mRNA includes 5′ UTRs, 5′ cap, 3′ UTRs, an open reading frame (ORF), and a poly(A) tail. Among them, 5′ cap consists of a negatively charged 5′, 5′-triphosphate bridge, and a positively charged base N7-methylguanosine (Figure 7), which directly affects the stability of mRNA. The 5′ position of the mRNA loses the free end of the phosphate group with the addition of Cap-0, which prevents degradation by alkaline phosphatase. Meanwhile, methyl groups of the two nucleotides following CAP-1 and Cap-2 block the 2′ OH groups, making it very stable to RNase A, RNase T1, and RNase T2 [124,125].

The optimization of the mRNA nucleotide composition should be prioritized to ameliorate the stability of mRNA–LNP vaccines. RNA conjugation refers to the covalent coupling of RNA to lipids, galactose, proteins/peptides or aptamers to prolong the circulation time in the body and increase the accumulation and uptake in specific tissues and cells. For example, the formation of RNA conjugates not only protects nucleic acids from degradation, but also enhances the binding to serum proteins [126]. Simultaneously, the structural modifications of the mRNA molecule are specifically designed to retain stability and increase the translation of the target antigen in vivo [124].

Cryo-TEM, small-angle X-ray scattering (SAXS), and small-angle neutron scattering (SANS) are usually used to measure the structure of mRNA–LNPs. Using this method, Arteta et al. demonstrated that LNPs have a disordered vertical and horizontal hexagonal internal structure with a characteristic distance of approximately 6 nm when mRNA is present, whereas LNPs without mRNA do not present this structure [76]. They also observed that DSPC and PEG-lipids, as well as some ionizable cationic lipids and cholesterol, are located on the surface of LNPs. Ionizable cationic lipids, cholesterol (depending on its concentration), water, and mRNA are mainly present in the nucleus. They demonstrated that mRNA is located in the water column surrounded by cationic lipids [76]. This suggests that the partial structural exposure of mRNA to water inside the LNP is one of the reasons why mRNA is unstable when stored in non-frozen conditions. Lyophilization is a process by which aquatic products transform into a solid state at low temperatures, and then remove water by sublimation in vacuum. This is a relatively mild drying method that can improve the stability of nanoparticles [127]. The lyophilized mRNA–LNPs can increase the stability and extend the shelf life of the product, so that it can be stored at 4 °C or at room temperature for a long period of time [128]. In the form of lyophilization, the mRNA–LNP vaccine can be easily transported to all parts of the world without cooling or freezing. However, these products have not been approved for marketing at present, and most of them are in the laboratory research stage. Studies have shown that the mRNA–LNP platform can also retain its stability by freeze-drying [129]. Muramatsu et al. inspected the stability of mRNA–LNPs by lyophilization at −80 °C, −20 °C, 4 °C, 25 °C, and 42 °C, respectively. It was observed that the product could be stored at room temperature for 12 weeks and stored at 4 °C for 24 weeks. The physical and chemical properties of each component did not change significantly [130]. However, with the increase in the storage temperature, mRNA degradation was more obvious. It is worth noting that the freeze-dried mRNA vaccine for CMV infection independently developed by Moderna has entered the phase III clinical stage, and the therapeutic effect is remarkable. Therefore, lyophilization will be one of the important development methods used to improve the stability of mRNA–LNP vaccines in the future.

## 4. Conclusions

The landscape of treatment, diagnostics, and vaccines for COVID-19 is advancing at an alarming rate with promising results. mRNA vaccines have obvious advantages when dealing with virus mutations at a high efficiency, rapidly, and safely due to the short period of time required from the performing the design sequence to the clinical trial. The mRNA vaccine has played a pivotal role in helping humans to develop immunity against COVID-19 and in preventing the spread of the virus. It is worth emphasizing that LNPs have various advantages over other carriers, including the protection of mRNA against degradation, co-delivery with adjuvants, and the ease of simple synthesis. At present, researchers have conducted in-depth research on LNPs using lipid carriers to deliver drugs, and have gained considerable experience in raw material control, process development, carrier-quality standards, and safety evaluation. Effective LNP-delivery technology is not only critical for the development of new mRNA vaccines, but also has the potential to deliver mRNA, gene, and CRISPR gene-editing therapies. DDSs mediated by LNPs, LPNs, lipoplexes, or LLPs is a field that is clearly progressing. It is expected that very soon new formulations will achieve clinical trials or enter the market of transfection reagents, especially in clinical applications.

Although mRNA–LNP vaccines are being developed at an unprecedented rate, some problems and challenges remain. The main challenges faced by mRNA–LNP vaccines are as follows: (i) large-scale production; (ii) ultra-low-temperature transportation and storage of vaccines; (iii) the delivery efficacy of mRNA; and (IV) strong immunogenicity, triggering an unnecessary immune response. The cellular uptake and endosomal escape of mRNA were enhanced by the rational lipid design [131,132,133]. However, it remains unclear how the ionizable cationic lipids in LNPs interact with mRNAs. Additionally, more research is required to confirm the internal structure of LNPs. Due to the instability of mRNA, the identification of impurities that may disrupt mRNA translation and negatively impact the activity of LNP-formulated mRNA products are prevalent. Furthermore, these impurities make it difficult to assess mRNA integrity in mRNA–LNPs using conventional techniques, such as CE. RP-IP HPLC provides specificity and sensitivity for detecting lipid–mRNA adducts [134]. The study of the interaction mechanism between LNPs and mRNA, improving the detection means of lipid–mRNA adducts, will play a critical role in improving the quality of mRNA–LNP vaccines.

mRNA is sensitive and easily degradable, requiring the use of ultra-low-temperature cold chains to ship COVID-19 vaccines around the world. Because cold-chain transportation is not available in many areas affected by COVID-19, a vaccine that can be stored at room temperature is highly desirable. Consequently, the major problem that, at present, needs to be resolved is the low stability and easy degradation of mRNA molecules. The good quality of mRNA vaccines relies on optimizing stability and delivery systems, which can be approached by rational molecular designs. Certainly, suitable additives and excipients can be selected to achieve transport stability at higher temperatures. In addition to alternative lipids and excipients, other formulation techniques, such as lyophilization, can be utilized to achieve better stability.

The emergence of variants of COVID-19 due to evolution and adaptation processes may encourage greater demands for the development of mRNA–LNP vaccines. The key to next-generation mRNA vaccine development is LNP and genome editing. In the future, mRNA–LNP vaccines may enable the adjustment of antigen designs and even permit the combining of sequences from several variants to respond to new mutations of the COVID-19 genome. The research reported a circular RNA (circRNA) vaccine that elicited potent neutralizing antibodies and T-cell responses by expressing the RBD of the spike protein, providing effective protection against SARS-CoV-2 in both mice and rhesus macaques [135]. New delivery system discoveries should help to meet the quality requirements that demand a balance between effectiveness, toxicity, and technology, aiming for market approval.

In summary, an excellent mRNA–LNP vaccine requires professional knowledge, a complete production system, and related technical teams from the selection and design of antigens, the selection of vectors, the preparation of samples, and the stable storage of vaccines. The ameliorated delivery efficiency of LNPs and their reduced toxicity, the optimized effective delivery of the carrier for mRNA, and the improved storage conditions of the product will be the subjects of our research in the future.

## Figures and Tables

**Figure 1 polymers-14-04195-f001:**
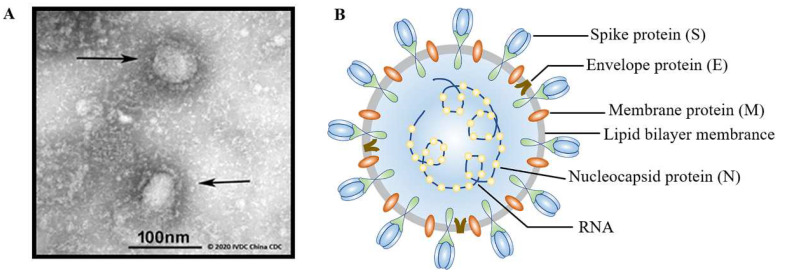
(**A**) Transmission electron microscopy (TEM) image of the COVID-19 virus (courtesy of IVDC, the Chinese Centre for Disease Control & Prevention; source: GISAID, https://www.gisaid.org/ (accessed on 21 October 2021). (**B**) Schematic representation of SARS-CoV-2 virus structure.

**Figure 2 polymers-14-04195-f002:**
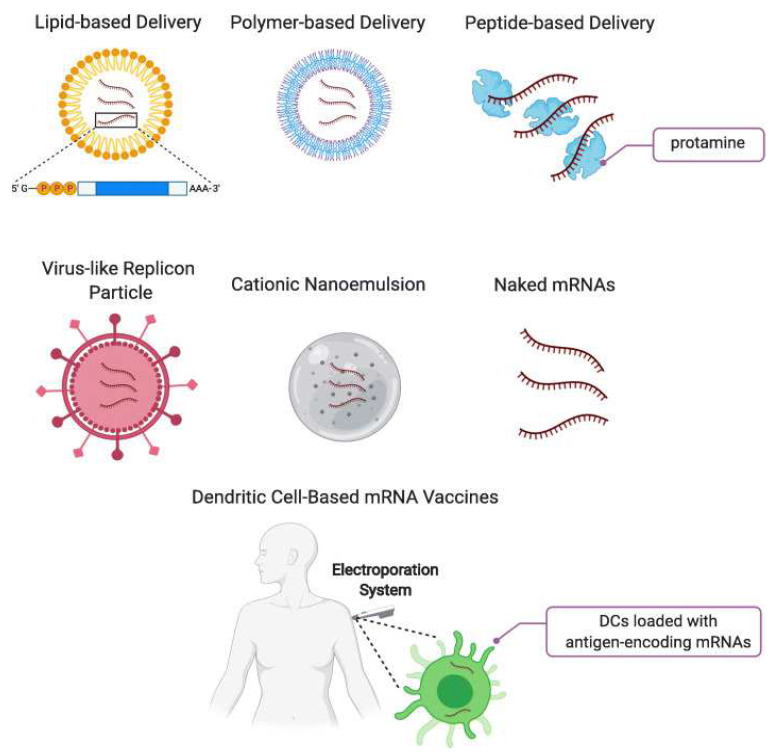
Summary of delivery methods for mRNA vaccines. Reprinted with permission from [55].

**Figure 3 polymers-14-04195-f003:**
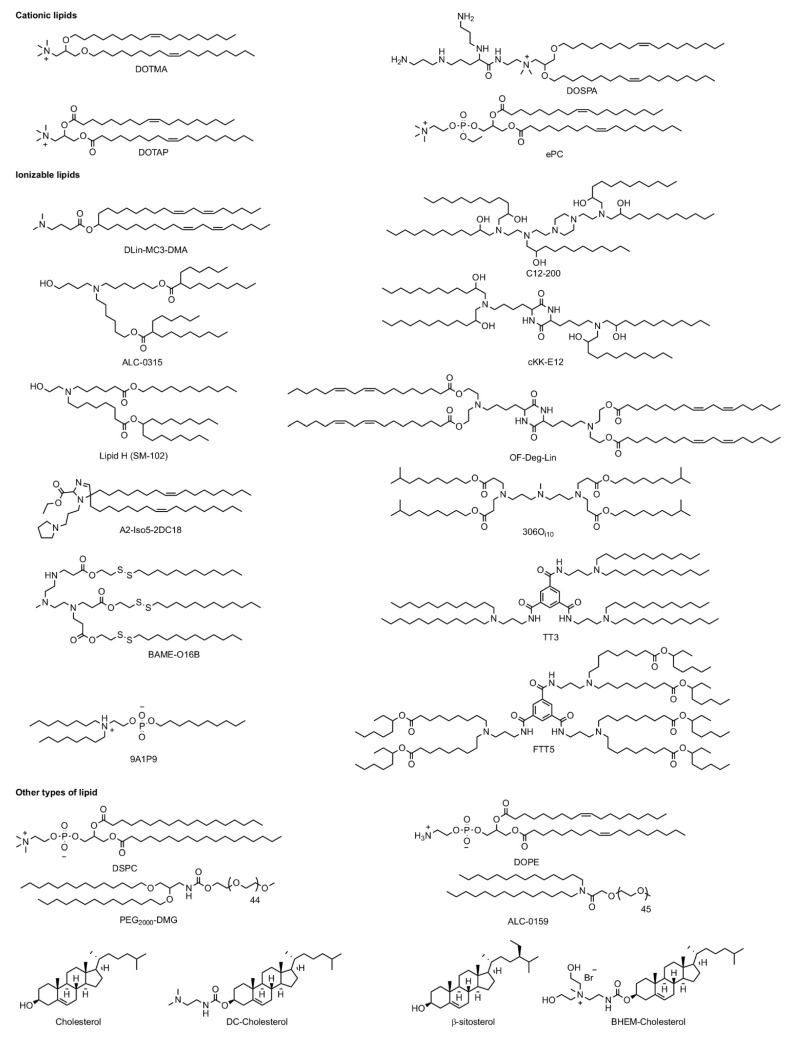
Chemical structures of lipids and lipid derivatives used for mRNA delivery. Reprinted with permission from [65].

**Figure 4 polymers-14-04195-f004:**
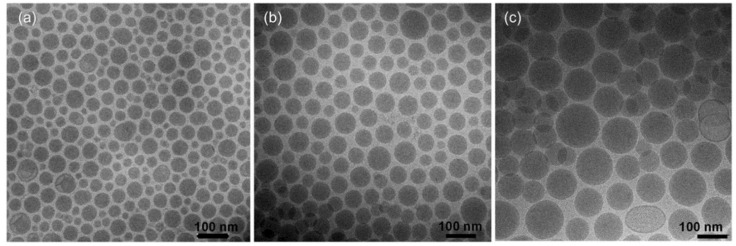
Cryo-TEM images of LNPs composed of DLin-MC3-DMA/DSPC/Chol/DMPE-PEG2000 in the ratio of 50:10:40-X:X for X:3 (**a**), 1.5 (**b**), and 0.5 (**c**). Reprinted with permission from [76].

**Figure 5 polymers-14-04195-f005:**
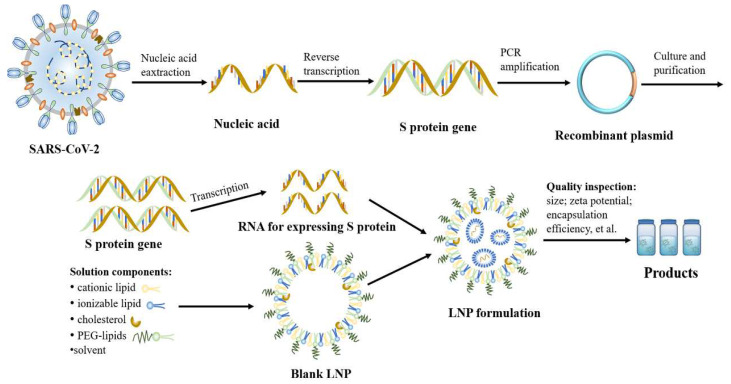
The workflow of mRNA–LNPs.

**Figure 6 polymers-14-04195-f006:**
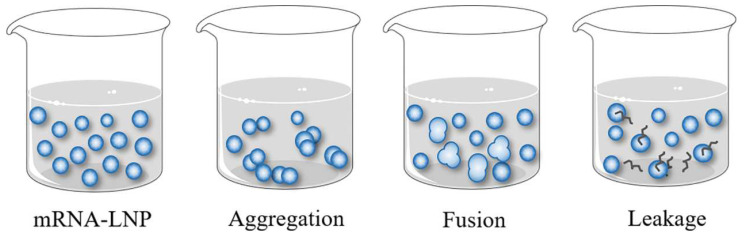
Physical instability mechanisms of mRNA–LNPs.

**Figure 7 polymers-14-04195-f007:**
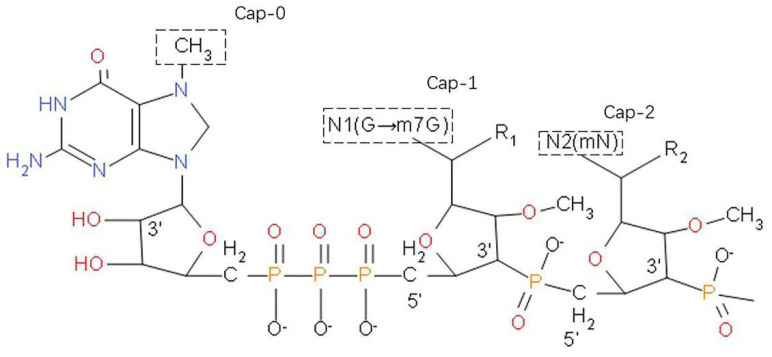
The 5′ cap of a eukaryotic mRNA chemical structure. Cap-0: add methyl to the 7-position of terminal G so that it has the cap of a single methyl group. Cap-1: add a methyl group to the second-base sugar chain 2′-O position. Cap-2: the 2′-O position of the third-base sugar chain will be methylated in the presence of Cap-1. Reprinted with permission from [124].

**Table 1 polymers-14-04195-t001:** LNP-based mRNA vaccines.

Indication	Target	LNP Composition	Administration Route	Animal Model	Company	Reference
Influenza virus	Full-length hemagglutinin (HA) of H10N8	DSPC: Chol: PEG-lipid: proprietary lipid: GLA (9.83:38.5:1.5:50:0.17)	i.m./i.d.	NHP	Moderna	[13]
Influenza virus	Full-length HA	DSPC: Chol: PEG-lipid: proprietary lipid (10:38.5:1.5:50)	i.m./i.d.	Mice, rabbits, ferrets	BioNTech	[14]
Influenza virus	H10 HA	DSPC: Chol: PEG-lipid: proprietary lipid (10:38.5:1.5:50)	i.m./i.d.	NHP	Moderna	[15]
Zika virus infection	ZIKV prM-E proteins	DSPC: Chol: PEG-lipid: proprietary lipid (10:38.5:1.5:50)	i.m.	Mice	Moderna	[16]
Zika virus infection	ZIKV prM-E proteins	DSPC: Chol: PEG-lipid: proprietary lipid (10:38.5:1.5:50)	i.d.	Mice, NHP	BioNTech	[17]
H7N9 influenza virus	SAM-encoding influenza H1 HA antigen obtained from the H1N1 virus	DSPC: Chol: PEG2000-DMG:DLinDMA (10:48:2:40)	i.m.	Mice	Moderna	[18]
H10N8 and H7N9 influenza virus	SAM-encoding HA proteins of H10N8 or H7N9	DSPC: Chol: PEG-lipid: proprietary lipid (10:38.5:1.5:50)	i.m./i.d.	Mice, ferrets, NHP	Novartis Vaccines and Diagnostics	[19]
Respiratory syncytial virus	SAM-encoding respiratory syncytial virus fusion glycoprotein	DSPC: Chol: PEG2000-DMG:DLinDMA (10:48:2:40)	i.m.	Mice	Novartis Vaccines and Diagnostics	[20]
Chikungunya infection	Human antibody CHKV-24	DSPC: Chol: PEG-lipid: proprietary lipid (10:38.5:1.5:50)	i.v.	Mice, NHP	Moderna	[21]
Toxoplasma gondii infection	SAM-encoding NTPase-II antigen	DSPC: Chol: PEG2000-DMG: DLinDMA (10:48:2:40)	i.m.	Mice	Wenzhou Medical University	[22]

**Table 2 polymers-14-04195-t002:** Information about marketed mRNA–LNP drug products for COVID-19 [6,23,24].

Company	Moderna	BioNTech
Product name	mRNA-1273	BNT162b2
mRNA type	Nucleoside modified mRNA	Nucleoside modified mRNA
Route of administration	Intramuscular	Intramuscular
mRNA dose	100μg	30 μg
mRNA encoding	Spike protein	Spike protein
Ionizable cationic lipid	SM-102	ALC-0315
Helper lipids	DSPC; cholesterol	DSPC; cholesterol
PEGylated lipid	PEG2000-DMG	ALC-0159
Molar lipid ratios (%) Ionizable cationic lipid: neutral lipid: cholesterol: PEGylated lipid	50:10:38.5:1.5	46.3:9.4:42.7:1.6
Clinical trials. Gov identifier	NCT04470427	NCT04368728
Overall protection rate	94.1%	95.0%

## Data Availability

Not applicable.

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
