# Peer review of "Design Strategies for and Stability of mRNA–Lipid Nanoparticle COVID-19 Vaccines"

_polymers, 2022, doi:10.3390/polym14194195_

Round 1

Reviewer 1 Report

In my opinion, given the vast literature and the very high number of reviews on mRNA-LNPs, Authors need to cover an angle not discussed by others for their review to have an interest; I would propose to focus on structure.

Authors need to discuss other types of lipid-based mRNA DDS such as hybrid biopolymers, lipoplexes, lipopolyplexes and nanocapsules.

Authors need to summarize SAXS/SANS/cryoTEM data on ionizabe and helper lipids. A paragraph on lipid organization should be added.

Authors need to discuss the mRNA-LNPs vaccines under devbelopment to tackle SARS-CoV2 variatns.

Authors need to explain what they mean by "LNP cannot complex nucleic acids".

Ref 61 and 62 are not LNPs stricto sensu.

Authors need to add a paragraph on biodegradability strategies (eg esters).

The claim of biocompatibility needs to be supported by data. Indeed, LNPs were shown to induce liver damage in animals and T cell hepatitis in humans (Alden et al, Curr Issues Mol Biol 2022; Boettler T et al J Hepatol 2022).

Authors need to add the structures of the PEGylated lipids used and particularly explain the choice for different PEGs in Moderna and Biontech LNPs;

In 2.3, the different strategies developped for large-scale production of mRNA need to be presented and detailed (eg Invitrogen's Dynabeads). Auhtos need to present the methods for mRNA purification and quality check, Authors should discuss a recent review: https://dx.doi.org/10.3390/vaccines9010003

Authors need to discuss lipid-RNA adducts: https://doi.org/10.1038/s41467-021-26926-0

Reviewer 2 Report

The manuscript gave an overview on the formulations of lipid-nanoparticles mRNA vaccine against COVID-19 and the current methods to improve vaccine stability. The research field of LNP/mRNA vaccines is rapidly growing and the in-depth knowledge of the rational and technologies to enhance LNP/RNA stability is needed and will be beneficial to public health interventions. However, there are few major issues

1.            the manuscript was written in poor English language.  Rewriting is needed throughout for better clarity.

2.            In the introduction, authors discussed about the therapeutic and preventive mRNA vaccine and summarized into table-1. In the following sections, authors discussed the aspects of the general methods to formulate mRNA/LNP formulation including antigen selection, mRNA and LNP preparations. those information were essential for vaccine technology to make preventive vaccine. There was no information provided for therapeutic vaccine. Author shall clarify the aim of this manuscript. In addition, Table-1 doesn’t correlate well to the COVID-19 vaccine. 

3. Overall, the manuscript discussed the general aspects of mRNA/LNP technology against COVID-19, which are wildly known to general public. However, there are many COVID-19 specific preclinical studies that are underdevelopment using different strategies, including mRNA modification, RBD vs full S protein, different delivery carrier formulations, route of delivery and extra. The immune outcomes are determined by various approach used. In my view, authors shall expend into in-depth reviewing of those technologies and bring up the scientific insights into this review article, in order to be more attractive to LNP/mRNA field.

Reviewer 3 Report

In the present manuscript entitled “Design strategies and stability of mRNA-lipid nanoparticle COVID-19 vaccines’’ authors have compiled the different points which should be considered in design strategies and physical and chemical stability of the vaccines. This topic is already widely covered through various reports and I don’t see any addition to the existing literature by this report. A few previously published almost similar reports are mentioned here for your reference (https://www.sciencedirect.com/science/article/pii/S0378517321003914, https://www.nature.com/articles/s41578-021-00358-0, https://www.mdpi.com/2076-393X/9/9/1033, https://www.sciencedirect.com/science/article/abs/pii/S1879625721000304, https://www.sciencedirect.com/science/article/abs/pii/S167385272100045X). In short, novelty is missing in the current article.

Round 2

Reviewer 1 Report

Authors adressed all comments, I fully support publication of the revised paper.

Author Response

We thank the reviewer very much for his/her positive comments.

Reviewer 2 Report

1.                English writing has been improved. However, issues in text body with grammar, spelling, punctuation, need to be checked again for better clarity.

2.                Information in table-2 need to be confirmed, and reference is needed. Moderna spike protein is not prefusion stablised? Also, should ALC-0159/ PEG-DMG be the components of BioNtech and Moderna, respectively? The molar lipid ratio for each product need to be confirmed as well.

3.                What is GISAID? The full name need to be placed in the first place where it present.

4.                In page-7,  sentences, quote “COVID-19 can activate innate and adaptive immune responses. The S protein of the new coronal coronavirus is a key protein that combines the virus and host cells. Therefore, many vaccines are currently in development, at present, target the S protein as the main antigen”, make no sense, and thus need to re-phases.

5.                References are needed in page-7 where the sentence quote “For example, LNPs can effectively encapsulate mRNA for effective and efficient delivery into a cell. Moreover, LNPs consists of biocompatible materials suitable for human use, which can be synthesized in at a large scale under the GMP level. More importantly, LNPs areis robustly synthesized, and its their components can be easily adjusted to increase cellular uptake and reduce cellular toxicity”.

6.                figure-4 on page11 need to be moved after fig-3.

7.                Fig-3 list a number of lipids that were used. However, more clarification is needed for example they were used in preclinical study? Clinical? Or authorized?

8.                In page-10, references are needed for sentences where quote “mRNA-–LNP formulations containing PEG2000-DMG have a higher delivery efficacy in vivo than formulations containing ALC-0159. Moreover, the separation of PEG2000-DMG from LNP is quicker than that of ALC-0159, facilitating the cellular uptake and endosomal escape of LNP”,.

9.                In Page-13, what is DCS?  What “via RNA-LPX” mean?

10.             In Page-15, what is “RNP”?

11.             In page-15, sentence quote” Specifically, in order to obtain nanoparticles with a uniform particle size and good polydispersity index (PDI), the flow rates of the aqueous and the organic phases is were 1.5 ml/min, and 0.5 ml/min, respectively” need to be moved to page16, following the end of first paragraph.

12.             In page-16, sentence quote” Plasmids and lipid excipients are essential in the process of mRNA vaccine research. In the whole entire quality control process…… should also be considered”. Although Plasmid is important, I think the author want to say that it is the mRNA molecules and lipid excipients are essential …. Plasmid is the starting material to produce mRNA, it is actually the quality of mRNA that determine the potency of vaccine.

13.             In page-17, more clarification is needed for the sentence quote “However, the vaccine can be stored up to 30 days in at 2℃ to 8℃”. Otherwise it shall be removed.

14.             In page-20, the formation of RNA conjugates was mentioned. It is better to clarify what RNA conjugation was.

15.             In page-20, reference is need for the statement quote” Simultaneously, the structural modifications of the mRNA molecule are specifically designed to keep retain stability and increase the translation of the target antigen in vivo”.

Reviewer 3 Report

Responses are satisfactory.

Author Response

(The authors gave the same response as above.)
